# Comparative Genomics of *Beggiatoa leptomitoformis* Strains D-401 and D-402^T^ with Contrasting Physiology But Extremely High Level of Genomic Identity

**DOI:** 10.3390/microorganisms8060928

**Published:** 2020-06-19

**Authors:** Tatyana S. Rudenko, Sergey V. Tarlachkov, Nikolay D. Shatskiy, Margarita Yu. Grabovich

**Affiliations:** 1Department of Biochemistry and Cell Physiology, Voronezh State University, Universitetskaya pl., 1, Voronezh 394018, Russia; ipigun6292@gmail.com (T.S.R.); kolyan9572@mail.ru (N.D.S.); 2All-Russian Collection of Microorganisms (VKM), G.K. Skryabin Institute of Biochemistry and Physiology of Microorganisms, Pushchino Scientific Center for Biological Research of the Russian Academy of Sciences, Federal Research Center, pr. Nauki, 5, Pushchino 142290, Russia; sergey@tarlachkov.ru; 3Branch of Shemyakin and Ovchinnikov Institute of Bioorganic Chemistry of the Russian Academy of Sciences, pr. Nauki, 6, Pushchino 142290, Russia

**Keywords:** *Beggiatoa leptomitoformis*, comparative genomics, transposon, dissimilatory sulfur metabolism, lithotrophic growth, sulfur globule protein

## Abstract

Representatives of filamentous colorless sulfur-oxidizing bacteria often dominate in sulfide biotopes, preventing the diffusion of toxic sulfide into the water column. One of the most intriguing groups is a recently described *Beggiatoa leptomitoformis* including strains D-401 and D-402^T^. Both strains have identical genes encoding enzymes which are involved in the oxidation of hydrogen sulfide and thiosulfate. Surprisingly, the *B. leptomitoformis* strain D-401 is not capable to grow lithotrophically in the presence of reduced sulfur compounds and to accumulate elemental sulfur inside the cells, in contrast to the D-402^T^ strain. In general, genomes of D-401 and D-402^T^ have an extremely high level of identity and only differ in 1 single-letter substitution, 4 single-letter indels, and 16 long inserts. Among long inserts, 14 are transposons. It was shown that in the D-401 strain, a gene coding for a sulfur globule protein was disrupted by one of the mentioned transposons. Based on comparative genomics, RT-qPCR, and HPLC-MS/MS, we can conclude that this gene plays a crucial role in the formation of the sulfur globules inside the cells, and the disruption of its function prevents lithotrophic growth of *B. leptomitoformis* in the presence of reduced sulfur compounds.

## 1. Introduction

Colorless sulfur bacteria were discovered by S.N. Winogradsky at the end of the 19th century. Using *Beggiatoa* as an example, S.N. Winogradsky formulated the idea of chemosynthesis (chemolithoautotrophy) which would become a key to understanding the metabolism and biodiversity of bacteria and to developing a concept of the role of microorganisms in the biosphere.

Representatives of the genus *Beggiatoa* are typical inhabitants of sulfide environments. These motile microorganisms form mats on the surface of sulfide sediments and are located in the layer where inorganic oxidizing and reducing agents are encountered. They are typical gradient organisms and are able to use reduced sulfur compounds as electron donors for energy metabolism. Such oxidation processes are accompanied by an accumulation of elemental sulfur within cells [1,2]. Currently the genus *Beggiatoa* includes two species with valid names for which pure cultures are available—*Beggiatoa alba* B18LD^T^ (ATCC 33555^T^), *Beggiatoa alba* B15LD (DSM 1416), *Beggiatoa leptomitoformis* D-402^T^ (DSM 14946^T^), and *Beggiatoa leptomitoformis* D-401 (DSM 14945^T^).

Surprisingly, *B. leptomitoformis* strains D-402^T^ and D-401 possess metabolic and morphological differences in the presence of hydrogen sulfide and thiosulfate. It was found that strain D-401 was not capable of lithotrophic growth [3] and did not accumulate elemental sulfur inside cells, while strain D-402^T^ was capable of to lithoheterotrophic and lithoautotrophic growth, and in the presence of reduced sulfur, compounds deposited an abundant amount of elemental sulfur inside the cells which can reach 20 to 50% of dry weight [4].

The complete genomic sequences of *B. leptomitoformis* strains D-401 and D-402^T^ were obtained in 2015 and 2018 [5,6]. Despite the fact that these strains were isolated from geographically remote habitats, the similarity of their genomes is more than 99.5% [5]. Therefore, elucidation of the molecular, biochemical, and evolutionary mechanisms responsible for such remarkable phenotypic differences with high genomic similarity is of great interest. This will contribute to the understanding of the evolution and ecology of the family Beggiatoaceae.

We studied the differences between genomes of strains D-401 and D-402^T^ and determined the role of these differences in sulfur metabolism connected with the ability to lithotrophic growth and accumulation of elemental sulfur in the presence of reduced sulfur compounds.

## 2. Materials and Methods

### 2.1. Cultivation

Medium for lithoheterotrophic cultivation of the *Beggiatoa leptomitoformis* strains contained the following composition: (g × L^−1^): NaNO_3_, 0.62; NaH_2_PO_4_, 0.125; CaCl_2_ × 2H_2_O, 0.3; Na_2_SO_4_, 0.5; KCl, 0.125; MgCl_2_ × 6H_2_O, 0.5; NaHCO_3_, 0.125; peptone, 0.2; lactate, 0.2; Na_2_S_2_O_3_ × 5H_2_O, 1.0. The pH of the medium was adjusted to 7.0 before inoculation. Before inoculation, the medium was supplemented with vitamins and microelements [7]. Thiosulfate was not added to the medium for organoheterotrophic cultivation; for lithoautotrophic growth, organic compounds were not added and the concentration of NaHCO_3_ was increased to 0.5 g × L^−1^. Cultures were grown at 27 °C.

To create microaerobic conditions, 0.5 L bottles with polybutyl rubber gaskets and screwed metal caps were filled with 50 mL freshly boiled medium and purged with argon. After that, the required volume of air was introduced into the bottles: the final concentration of O_2_ in the gas phase was 2.0% [3].

### 2.2. Genome Polishing

DNA was isolated using the FastDNA™ SpinKit (MP Biomedicals, Irvine, CA, USA). The libraries were prepared using the KAPA HyperPlus kit with adapters from the KAPA Dual-Indexed Adapter Kit. The resulting libraries were sequenced on the Illumina MiSeq platform with 2 × 300 bp reads. Raw sequencing data preprocessing was performed using Trimmomatic v. 0.39 [8], and then the resulted reads were mapped to the corresponding genomic assemblies using Bowtie2 v. 2.3.4.3 [9]. Genome corrections were performed using Pilon v. 1.23 [10].

### 2.3. Genome Analysis

The MEGA 6 program was used [11] for cluster analysis of long inserts. Sequence alignment was performed using ClustalW [12]. Trees were constructed by the neighbor-joining method [13] using the *p*-distance. Confidence values of branch nodes were evaluated using bootstrap analysis by constructing 1000 alternative trees. Homologous protein sequences were searched using the NCBI database (https://blast.ncbi.nlm.nih.gov/Blast.cgi). A search for conserved domains in the protein sequences was performed using the NCBI database (https://www.ncbi.nlm.nih.gov/Structure/cdd/wrpsb.cgi). A search of signal peptides in amino acid sequences was performed using the SignalP service (http://www.cbs.dtu.dk/services/SignalP).

### 2.4. Gene Expression Analysis

RNA was isolated using the ExtractRNA reagent (Evrogen, Moscow, Russia) in accordance with the manufacturer’s protocol. RNA quality was evaluated by electrophoresis on a 2% agarose gel with 2.2 M formaldehyde added. RNA concentration was measured using an HS Qubit RNA assay kit (Thermo Fisher Scientific, Waltham, MA, USA) on a Qubit 2.0 fluorometer (Thermo Fisher Scientific, Waltham, MA, USA). Then, 1000 ng of RNA was reverse transcribed using M-MulV (SybEnzyme, Moscow, Russia) according to the manufacturer’s protocol. Quantitative RT-PCR was performed using SYBR Green I on a Bio-Rad CFX96TM real-time system (Bio-Rad, Hercules, CA, USA).

In order to find optimal amplification conditions, a temperature gradient was used. The final program included 95 °C, 3′ + ((95°, 20″ + 60 °C, 20″ + 72 °C, 30″) × 39). Fragments of genes encoding the proteins ALG67575.1, ALG68134.1, and ALG67202.2 were amplified using primers 5′-GGGGCAACAATGAATGGGGT-3′ and 5′-GACGACCATAACCACGACCG-3′ for ALG67575.1, 5′-GCTGAAAAATGGGACGCAGG-3′ and 5′-TCACCCACGTTGCCATAACC-3′ for ALG68134.1, 5′-CTTTGTGATTTGGGCGGCAA-3′ and 5′-AGATGCGACAAGTGATGCGG-3′ for ALG67202.2; all primers were designed with PrimerBLAST (http://www.ncbi.nlm.nih.gov/tools/primer-blast).

### 2.5. Enzymatic Assay

Sulfite oxidase activity (SOE) (EC 1.8.5.6) was measured in a buffer containing 50 mM Tris-HCl pH 8.0, 2 mM ferricyanide, 2 mM sodium sulfite, EDTA 2.9 mM. The activity was determined spectrophotometrically at 420 nm by a decrease in optical density as a result of ferricyanide reduction [14].

The protein concentration in the sample for the measurement of the enzyme activity was in the range of 100–500 μg mL^−1^ (crude cell extract).

### 2.6. Sample Preparation for Proteomic Analysis

Cells were resuspended in 200 μL of lysis buffer containing 1% dithiothreitol, 4% CHAPS, 7M urea, 2M thiourea, and 5% ampholytes 3/10. Lysate was prepared using a Potter homogenizer. The homogenate was centrifuged at 5000× *g* for 10 min at 20 °C and the supernatant was collected. Protein concentration was determined by the Bradford method using a commercial set Coomassie Protein Assay Reagent, Thermo Fisher according to the manufacturer’s instructions.

### 2.7. Two-Dimensional Gel Electrophoresis (2-DE)

2-DE was performed as described in [15] on a PROTEAN II xi 2-D Cell system (Bio-Rad, Hercules, CA, USA).

When conducting isoelectric focusing, the pH gradient was from 3 to 10 (servalites, Serva Electrophoresis GmbH, Heidelberg, Germany). The amount of the sample was 120 μg of protein per tube. Isofocusing was performed for 16 h under the following modes: 100 V-45 min, 200 V-45 min, 300 V-45 min, 400 V-45 min, 500 V-45 min, 600 V-45 min, 700 V-10 h, and 900 V-1.5 h.

Electrophoresis of the samples obtained after isoelectric focusing was carried out in a gradient acrylamide gel with sodium dodecyl sulfate (7.5–25%) at a voltage of 300 V. Before applying to the second dimension, the samples were incubated for 20 min in a solution containing dithiothreitol (6M urea, 2% sodium dodecyl sulfate, 10 mM dithiothreitol, 0.5M Tris-HCl, pH 6.8) to prevent oxidation of sulfhydryl groups in proteins. For visual analysis of the distribution of protein components and for mass spectrometric analysis, the gels were stained with Brilliant Blue R Staining Solution (Sigma, Georgetown, SC, USA).

The Infinity1000/26MX gel documentation system (VilberLourmat, Collégien, France) was used to obtain protein maps.

### 2.8. High Performance Liquid Chromatography in Combination with Tandem Mass Spectrometry (HPLC-MS/MS)

Proteins from the strains D-401 and D-402^T^ were subjected to trypsin (Promega, Madison, WI, USA) and chymotrypsin (Promega, Madison, WI, USA) digestion according to the FASP protocol on 10 kDa-MWCO filters (Microcon Centrifugal Filter 10 kDa, Millipore, Burlington, MA, USA), as described previously [16].

The peptides were separated with high-performance liquid chromatography (HPLC, Ultimate 3000 Nano LC System, Thermo Scientific, Rockwell, IL, USA) in a 15-cm long C18 column with an inner diameter of 75 μm (Acclaim PepMap RSLC, Thermo Fisher Scientific, Rockwell, IL, USA). The peptides were eluted with a gradient of buffer B (80% acetonitrile, 0.1% formic acid) at a flow rate of 0.3 μL min^−1^. Total run time was 90 min which included the initial 4 min of column equilibration to buffer A (0.1% formic acid), then the gradient from 5 to 35% of buffer B over 65 min, then 6 min to reach 99% of buffer B, flushing 10 min with 99% of buffer B, and 5 min re-equilibration to buffer A. MS analysis was performed in triplicate with a Q Exactive HF mass spectrometer (Q Exactive HF Hybrid Quadrupole-Orbitrap Mass spectrometer, Thermo Fisher Scientific, Rockwell, IL, USA). Mass spectra were acquired at a resolution of 120,000 (MS) and 15,000 (MS/MS) in a *m/z* range of 350–1500 (MS) and 100–2000 (MS/MS). An isolation threshold of 100,000 counts was determined for precursor’s selection and the top 10 precursors were chosen for fragmentation with high-energy collisional dissociation at 30 NCE and 100 ms accumulation time. Precursors with a charged state of +1 were rejected and all measured precursors were excluded from measurement for 20 s.

### 2.9. Proteomic Data Processing

The proteins of the strains were identified with the SearchGUI v. 3.3.17 program using simultaneous X!Tandem, OMSSA, and MS-GF+ algorithms [17] with protein sequences databases TrEMBL *Beggiatoa leptomitoformis* D-401 (3445 entries) and *Beggiatoa leptomitoformis* D-402^T^ (3447 entries). Parameters were set as follows: no enzyme specificity; MS tolerance of 5 ppm and 0.01 Da tolerance for MS/MS ions; with carbamidomethylation of C as a fixed modification and oxidation of M as a variable modification. The SearchGUI output was analyzed and visualized in PeptideShaker [18] v. 1.16.44. Peptide-Spectrum Matches, peptides, and proteins were validated at a 1.0% False Discovery Rate estimated using the decoy hit distribution. Only proteins having at least two unique peptides were considered as positively identified.

## 3. Results

### 3.1. Improving Assembly Quality

In the present research, we use complete genomic sequences of *B. leptomitoformis* D-402^T^ and D-401, previously obtained using PacBio RSII platform [5,6]. Recent studies reported that PacBio RSII sequencing technology has a high error rate in homopolymer regions [19]. At the same time, our preliminary analysis of genomic sequences of strains D-402^T^ and D-401 showed a rather large number of homopolymer sites. Thus, to improve the quality of assemblies, we polished previously published sequences using Illumina reads obtained in the present research. Polished assemblies had been published in GenBank under accession numbers CP012373.2 and CP018889.2 for *B. leptomitoformis* D-402^T^ and D-401, respectively. In total, we corrected 65 sites for the genome of strain D-401 and 22 sites for strain D-402^T^. After correction, the number of pseudogenes decreased approximately by factor 2 due to the correction of the shifts of the reading frames.

The refined sequences of the complete genomes of *B. leptomitoformis* strains allowed us to conduct a comparative analysis of homologous genes encoding all the necessary enzymes involved in the oxidation of hydrogen sulfide, thiosulfate, elemental sulfur, and sulfite (Table 1). In particular, genes of periplasmic sulfide-oxidizing enzymes were found in the genomes of the strains D-401 and D-402^T^: sulfide:quinoneoxyderoductase (*sqrQ*) type I (*sqrA*) and type VI (*sqrF*); flavocytochrome c-sulfide dehydrogenase (FCSD) type *fccB*. Additionally, genes encoding enzymes that can be involved in the oxidation of thiosulfate due to the branched pathway were identified in two strains and SoxB activity was detected only in the strain D-402^T^ [20].

The mechanism of oxidation of sulfur to sulfite in representatives of the genus *Beggiatoa* is still not fully understood.

Oxidation of sulfite to sulfate by prokaryotes can occur in two ways: direct and indirect. Genes (*aprAB*, *sat*/*sopT*, *apt*) encoding the enzymes of the indirect pathway for the oxidation of sulfite to sulfate were not identified in any of the analyzed *B. leptomitoformis* genomes, and we did not find corresponding enzymatic activity [20]. The direct pathway for sulfite oxidation can be catalyzed by several enzymes: soluble periplasmic sulfite:ferricitochrome c oxidoreductase, cytoplasmic sulfite:quinone oxidoreductase SoeABC membrane-bound complex. The genes of both SorAB subunits were not identified in the genomes of *B. leptomitoformis* strains, while the genes encoding SoeABC were found and the enzymatic activity of the complex was shown (50–70 nmol × min^−1^ × mg protein^−1^) with lithoautotrophic growth of bacteria under microaerobic conditions.

It can be concluded that both strains carry the genes necessary for the dissimilation of sulfur metabolism. However, only strain D-402^T^ is capable of lithoheterotrophic and lithoautotrophic growth and an accumulation of elemental sulfur inside cells in the presence of thiosulfate or hydrogen sulfide (Figure 1) [20]. Strain D-401 is not able to use reduced sulfur compounds as an electron donor for energy metabolism, it shows only organoheterotrophic growth.

Moreover, a comparative analysis of the coding and promoter regions of these genes did not reveal any differences between strain D-402^T^ and D-401. It can be suggested that other genes associated with the dissimilation of sulfur metabolism can also be changed.

### 3.2. Comparison of Complete Genomes

In order to identify any genetic differences that cause a physiology contrast between the strains, we performed a whole-genome alignment. There are no large rearrangements in the genomes. However, we identified 16 long inserts: eight inserts were found in the genome of strain D-402^T^ and eight inserts were found in the genome of strain D-401. These inserts can be divided into several types by length: 10 inserts with length equal to 853 bp, four inserts with length equal to 782 bp, one insert with length equal to 378 bp, and one insert with length equal to 11 bp. We also found four one-letter indels and one one-letter substitution. Identified loci that distinguish genomes do not form clusters and are evenly distributed over the genomes (Figure 2, Appendix A).

### 3.3. Long Inserts

Cluster analysis showed that long inserts can be divided into three separate types (Appendix A). Within each type of sequence, a high percent identity (>97%) is shown. Interestingly, long inserts of the second type are found only in strain D-401. We found direct and inverted repeats at the ends of sequences of types 1 and 2 (Appendix A), identical within each type which indicates that these sequences are transposons. For sequences of type 3, repeats were not found.

A search for transposons by the exact match of the inverted repeats in the genome of strain D-402^T^ additionally revealed 34 sequences of the first type and 12 sequences of the second type, and for strain D-401 we identified 32 and 16 sequences, respectively. Among them, two transposons of greater length (1731 and 1660 bp) were identified, belonging to type 1 and 2, respectively. These sequences found in the genome of strain D-402^T^ are completely identical to the sequences found in the genome of strain D-401. The full-sized sequences of each type contain four protein-coding genes (Appendix A) which are various transposases and nucleases. This further confirms that these sequences are transposons. Thus, strains D-402^T^ and D-401 have at least two types of transposons (types 1 and 2), and the sequence of type 3 is not a transposon.

### 3.4. Candidate Genes

The one-letter substitution falls into the coding region of the gene which encodes the response regulator, but it does not change the amino acid sequence of the protein. Therefore, we did not consider this difference as significant. Single-letter indels lead to frameshifts in the genes encoding the following putative proteins: BamA/TamA family outer membrane protein, long-chain acyl-CoA synthetase, and transposase. We suppose that these genes do not play a significant role in the lithotrophy processes in *B. leptomitoformis*. Three transposons found in the genome of strain D-401 fall into coding sequences and can disrupt the functioning of corresponding genes.

The first and most likely gene can encode the protein of the sulfur globules envelope (ALG67575.1). In representatives of the family Beggiatoaceae, elemental sulfur is deposited in cytoplasmic invaginates and is surrounded by a specific protein envelope. A 15-kDa protein was found around sulfur inclusions in *Beggiatoa alba* B18LD, when strain cultivated in medium with sulfide [21,22].

This class of proteins generally does not contain markable conserved domains. However, the sequence of the found protein enriched with regularly spaced prolines contains a cleavable N-terminal peptide necessary for transport to the periplasmic space. These features correspond to those for the previously described envelope proteins [23]. Similar genes were also found in phylogenetically close neighbors of *B. leptomitoformis* D-402^T^—*B. alba* B18LD and *Thioflexithrix psekupsensis* D3 (Appendix A).

The two other genes are hypothetical proteins. One of them can encode porin (ALG68134.1) which can form channels in the outer membrane through which small molecules, such as thiosulfate are able to penetrate. Thus, this gene can play a role in lithotrophic growth processes, acting as a transmembrane transporter. Another gene encodes a protein (ALG67202.2) containing a sensory domain with an unknown function.

To show the lack of full-sized proteins of the found genes in strain D-401, a proteomic analysis of the proteins of both strains which were cultivated in the presence of lactate and thiosulfate was performed. During the study of proteomic maps by the method of two-dimensional gel electrophoresis, differences in protein expression profiles were revealed (Figure 3). In strain D-401, none of the claimed proteins were found. We identified and analyzed proteins that corresponded to the calculated values of the molecular weights of the studied proteins without and with inserts. Mass-spectrometric analysis allowed us to identify peptides in the D-402^T^ sample corresponding to the full-size proteins of the sulfur globules envelope (ALG67575.1) and the hypothetical porin protein (ALG68134.1) (Appendix A). Protein ALG67202.2 was not identified in the sample of strain D-402^T^. The reason may be the low representation of the corresponding protein in the sample.

Additionally, for the analysis of proteins of interest, a more sensitive approach for proteomic analysis was used—high performance liquid chromatography in combination with tandem mass spectrometry. During the analysis of strain D-401 by HPLC-MS/MS, 847 proteins were reliably identified. When analyzing proteins of strain D-402^T^, 606 protein sequences were reliably identified, among which proteins ALG68134.1 (identifier in TrEMBL UPI0007069FE2) and ALG67575.1 (identifier in TrEMBL UPI00070625ED) were found.

Appendix A shows the protein sequences of candidate genes, where the peptides which were identified by MALDI mass spectrometry and HPLC-MS/MS (LC-MS/MS) are highlighted in bold and underlined, respectively. Appendix A shows that the protein ALG68134.1 is fairly well covered with peptides identified by HPLC-MS/MS which fully include the sequences of peptides detected by MALDI mass spectrometry. The sequence coverage of this protein is 71.6%. For protein ALG67575.1 (Appendix A), the peptide in the N-terminal region was detected only by MALDI mass spectrometry. The C-terminal region of the protein is covered with peptides which were detected using both mass-spectrometric approaches, and the total protein coverage of ALG67575.1 is 40.4%.

### 3.5. Candidate Genes Expression

To determine the involvement of the proteins ALG67575.1, ALG68134.1, and ALG67202.2 in the oxidation processes of reduced sulfur compounds in the strain D-402^T^, we studied expression of genes coding for these proteins in the presence or absence of thiosulfate in the culture medium.

As a result of RT-PCR, it was shown that the presence of thiosulfate in the medium does not affect the expression of genes of proteins ALG67202.2 and ALG68134.1. In the case of the gene which encodes the sulfur globules envelope protein (ALG67575.1), it was found that the addition of thiosulfate to the medium increases gene expression by 10 times compared with the growth without thiosulfate (Figure 4).

## 4. Discussion

The inability of the *B. leptomitoformis* strain D-401, in contrast to the D-402^T^ strain, to grow lithotrophically in the presence of reduced sulfur compounds may suggest the existence of mutations in the genes related to dissimilatory sulfur metabolism. However, we did not find any differences in the genes of sulfur metabolism between strain D-401 and D-402^T^ in both coding and promoter regions. However, in strain D-401, the presence of a transposon in the gene that encodes sulfur globules envelope protein was found which leads to morphological and physiological differences between strains D-401 and D-402^T^.

For some sulfur bacteria capable of accumulating elemental sulfur intracellularly, electron micrographs clearly show that sulfur globules form in the periplasmic space [22,24]. In these organisms, sulfur globules are surrounded by a protein envelope, for example, in representatives of the genus *Thiothrix* [25], purple sulfur bacteria [21,26,27,28], *Beggiatoa* [29], and *Thiovulum* [30].

Today, little is known about the protein of the sulfur globules envelope in representatives of the genus *Beggiatoa*. The vast majority of what is known regarding sulfur globules envelope comes from studies of the purple sulfur bacteria of the Chromatiaceae family [21,26,27,28]. In *Allochromatium vinosum*, the sulfur envelope is represented by three proteins (SgpA, SgpB, SgpC) [31]. The sulfur envelope of *Thiocapsa roseopersicina* contains two proteins of 10.7 and 8.7 kDa, both of which are homologous to large and small proteins of *A. vinosum* [32]. The presence of a single protein of 18.5 kDa of the sulfur globule in *Isochromatium buderi* [33] and the 15-kDa protein in *Beggiatoa* was found when bacteria were grown in a medium supplemented with sulfide [22]. Today, there is no clarity regarding the subunit structure of the protein of the sulfur globules envelope from *I. buderi* and *Beggiatoa*.

Our proteomic data confirm the presence of a protein of 15 kDa which performs the function of a sulfur globules envelope in *B. leptomitiformis* D-402^T^ and confirm its absence in strain D-401. Moreover, for strain D-402^T^, we have shown that the addition of thiosulfate to the culture medium increases the gene expression of this protein by 10 times compared with the cultivation without thiosulfate. Thus, obtained data allow us to assume that the protein ALG67575.1 is strictly necessary for lithotrophic growth in the presence of reduced sulfur compounds. Our data are corresponding to experiments with the purple sulfur bacterium *Allochromatium vinosum* [31]. The sulfur envelope proteins in *A. vinosum* are encoded by three genes *sgpABC*. It was shown that *sgpBC* double mutant completely lost the ability to use H_2_S as an electron donor and accumulate elemental sulfur inside the cells.

Comparison of biotopes where *Beggiatoa leptomitoformis* D-401 and D-402^T^ strains develop shows that strain D-401 lives in silt sediments of treatment facilities (Yaroslavl Oblast, Russia), where the concentration of H_2_S is not high (0.5–1.0 mg × L^−1^), while strain D-402^T^ was isolated from sulfur mat formed on irrigation fields contaminated with residential and agricultural wastewater (Moscow Region, Russia), where the concentration of H_2_S is 2–5 mg × L^−1^.

The incorporation of transposon into the corresponding gene in D-401 led to the “shutdown” of the lithotrophy process in the presence of reduced sulfur compounds which, in turn, allowed D-401 to develop in a biotope with a lower H_2_S and other bioenergetics nutrient content (organic compounds) than the biotope of D-402^T^. 

It is considered that transposons can easily leave genes and move within the genome. However, traces of ’cutting out’ transposons were not revealed in the studied strains, which indicate genome stability of the strains D-401 and D-402^T^, and we have been observing their properties for more than 35 years.

Along with our research, further study of representatives of the genus *Beggiatoa* could have far-reaching implications in environmental microbiology and biogeochemical cycling.

## Figures and Tables

**Figure 1 microorganisms-08-00928-f001:**
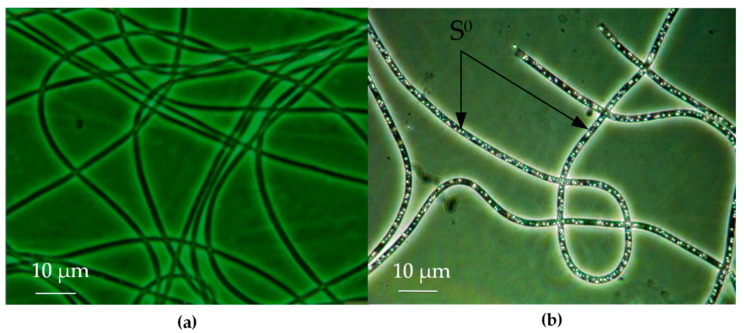
Phase contrast micrographs showing morphology of *Beggiatoa leptomitoformis* D-401 (**a**) and D-402^T^ (**b**) when growing on a medium with thiosulfate and lactate. Arrows indicate elemental sulfur (S^0^) inclusions in D-402^T^ cells.

**Figure 2 microorganisms-08-00928-f002:**
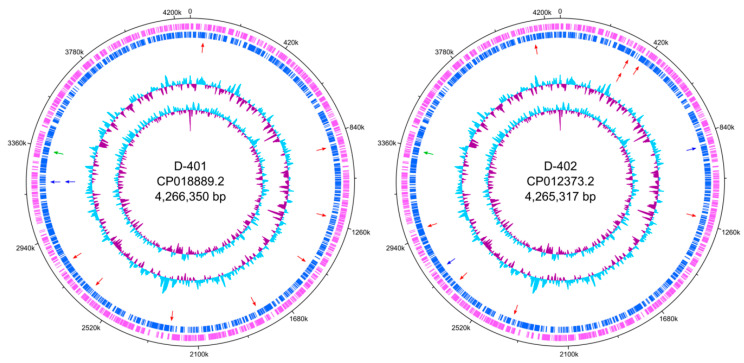
Map of differences between *B. leptomitoformis* D-401 and D-402^T^ genomes. Designations from the outer circle to the inner one: circle 1 and 2-predicted open reading frames in forward and reverse orientation, respectively, 3-long inserts (red), single-letter inserts (blue), single-letter substitution (green), 4-GC-plot, showing deviations from the mean value, 5-GC-skew. Coordinates are specified in bp.

**Figure 3 microorganisms-08-00928-f003:**
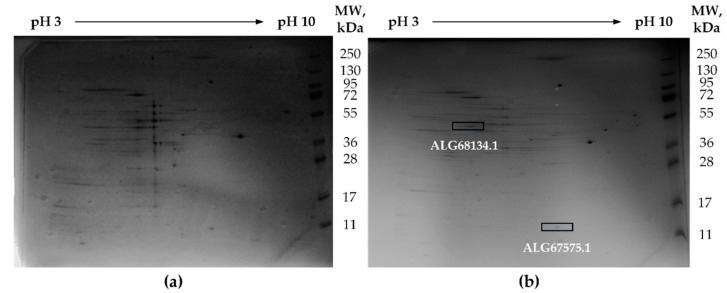
Protein expression profiles in *B. leptomitoformis* strains D-401 (**a**) and D-402^T^ (**b**) after two-dimensional gel electrophoresis on proteomic maps. Cultivation in the presence of thiosulfate and lactate.

**Figure 4 microorganisms-08-00928-f004:**
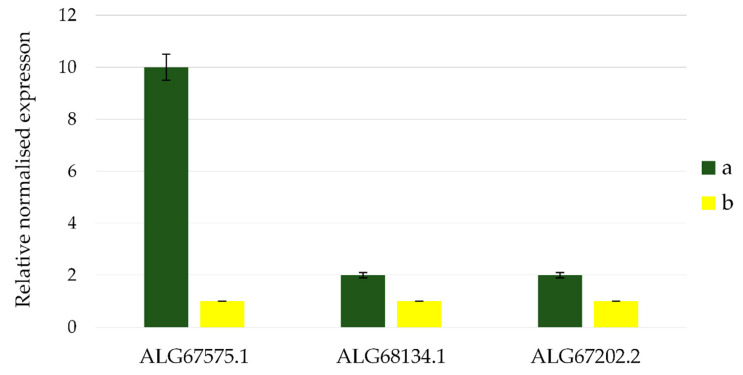
The expression of genes coding for ALG67575.1, ALG68134.1, and ALG67202.2 proteins of *B. leptomitoformis* strain D-402^T^ during lithoheterotrophic (**a**) and organotrophic (**b**) cultivation.

**Table 1 microorganisms-08-00928-t001:** Genes encoding the enzymes of dissimilation sulfur metabolism of *Beggiatoa leptomitoformis* D-401 and D-402^T^.

B. *leptomitoformis* D-401		B. *leptomitoformis* D-402^T^		Gene
Locus Tag	Protein Accession	Locus Tag	Protein Accession	
BLE401_14705	AUI69816.1	AL038_09565	ALG67913.1	*soxY*
BLE401_14710	AUI69817.1	AL038_09560	ALG67912.1	*soxZ*
BLE401_13150	AUI69545.1	AL038_11110	ALG68158.1	*soxAX*
BLE401_00760	AUI67367.1	AL038_05225	ALG67220.1	*soxB*
BLE401_03540	AUI67863.1	AL038_02505	ALG66791.1	*soxF*
BLE401_07405	AUI68548.1	AL038_16835	ALG69045.1	*sqr*
BLE401_16835	AUI70200.1	AL038_07480	ALG67570.1	*dsrE*
BLE401_16840	AUI70201.1	AL038_07475	ALG67569.1	*dsrF*
BLE401_16845	AUI70202.1	AL038_07470	ALG67568.1	*dsrH*
BLE401_16850	AUI70203.1	AL038_07465	ALG67567.1	*dsrC*
BLE401_14535	AUI70655.1	AL038_09735	ALG69423.1	*soeA*
BLE401_14540	AUI69788.1	AL038_09730	ALG67939.1	*soeB*
BLE401_14545	AUI69789.1	AL038_09725	ALG67938.1	*soeC*

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
