# Peer review of "Comparative Genomics of Beggiatoa leptomitoformis Strains D-401 and D-402T with Contrasting Physiology But Extremely High Level of Genomic Identity"

_microorganisms, 2020, doi:10.3390/microorganisms8060928_

Round 1
Reviewer 1 Report
Rudenko et al have the goal of elucidating the molecular and biochemical and evolutionary mechanisms responsible for a strong phenotypic difference between two strains of Beggiatoa. The phenotype involves the inability of one of the strains to accumulate sulfur. This was traced to a transposon insertion into a gene for a sulfur globule protein. The manuscript is focussed on the detection of differences between the two strains at the genetic, protein and RNA levels to account for the phenotype, which is a clearly visible difference in inclusions of sulfur within cells (viewed with the help of a microscope). Generally, I find the research to be interesting and well-performed.
Some concerns:
The authors have most likely found the cause for the phenotypic difference between the two strains, i.e., a transposon insertion into a gene important for sulfur accumulation. Why not remove all doubt and disrupt the gene in the sulfur accumulating strain to see if the mutation really does remove the ability to accumulate sulfur? Alternatively, or additionally, an intact copy of the gene could be re-introduced into the strain incapable of sulfur accumulation, to see if this returns the ability to accumulate sulfur. Can these bacteria be genetically manipulated?
Figure 1 provides a view of the phenotypic differences between the two strains. However, I could find no reference to Figure 1 in the main text, nor could I find any information on how the photo was generated, e.g., via phase contrast. The S and the black arrows also should be clearly explained in the figure legend.
Line 234 The authors write: ‘Single-letter indels also do not affect significant genes in strain D-401…’ I do not understand this statement. How can single letter indels not affect the reading frame of any gene?
Line 340 The authors state that this research along with further work could have far-reaching implications of wastewater treatement from hydrogen sulfide. I’m optimistic that this is correct, and I’d like to know what these implications are.
The English throughout the manuscript needs to be improved. Many sentences are barely understandable.
Reviewer 2 Report
General comments:
The manuscript Microorganisms-828530 entitled “Comparative genomics of Beggiatoa leptomitoformis 2 strains D-401 and D-402T with contrasting physiology 3 but extremely high level of genomic identity” by Rudenko et al. describes the analysis of two Beggiatoa leptomotoformis strains, D-401 and D-402, very closely related in terms of genome global identity but clearly distinct in term of reduced sulphur metabolism. Their genomic, RT-PCR and HPLC-MS/MS analysis revealed that a single transposon, in D-401, impacted the production of sulphur globule inside the cells and let the lithotrophic growth of this strain impossible, contrary to D-402. The manuscript described well done experiments. I think that authors are correct in their general crude analysis, except for Fig. 3 as detailed below, but in my point of view the comparison of both strains’ growth has to be compared and is missing in this work. I have also a concern about genomic evolution interpretation in the discussion of the data. Except these points, this manuscript is well written, with clearly exposed results and arguments. I consider therefore this work acceptable for publication in “Microorganisms”, with major amendments.
Specific comments:
-L188: Because the sulfite:ferricytochrome c oxidoreductase could be of SorAB or SorT types, I suggest to indicate the two types specifically or to use a common denomination “soluble periplasmic sulfite:ferricytochrome c oxidoreductase”. I also suggest, with the aim to be more informative, to precise “cytoplasmic sulfite:quinone oxidoreductase SoeABC membrane-bound complex” for the Soe.
Could the authors precise if neither of the SorA and SorB gene are retrieved in their strains or rather not both concomitantly? This is worth to precise because the SorT is devoid of cytc subunit.
-L192-197: oxidation of reduced sulphur compounds is a bioenergetics process as exposed by the authors themselves. The inability of D-401 to use these compounds should result in limited growth in the conditions described in Figure 1 (lithoautotrophic) compared to D-402T. I consider therefore that compared growth curves of D-401 and D-402T has to be shown, in addition to morphology of the cells. This data has not been published before and should be added to this work.
-L262: I do not follow the authors in this claiming. Fig. 3 Panel (b), shows in D-402T, a spot identified as ALG67202.2. In the text, however, the authors affirm that this protein is not identified in D-402T. The expression of the ALG67202.2 gene, shown in Fig. 4 is also in line with the presence of the protein, as shown in Fig. 3. The text has to be corrected. I suggest also to indicate the growth conditions of the cells (thiosulfate? Organoheterotrophic) having produced the protein material for experiments shown in Fig. 3
-L292: I suggest to precise, in the caption of Figure 4, that the data refer to D-402T.
-L328: figure 4 shows that expression of ALG67575 gene is not constitutive but rather inducible. I therefore consider that the simple presence of ALG67575 gene in its genome could not be detrimental to D-401 under sulfide or thiosulfate. I thus suggest not to present the shutdown of ALG67575 as beneficial for the strain in a biotope with a low concentration of H2S but rather to present the shutdown of ALG67575 as having led the D-401 shift to a biotope with less H2S and with other bioenergetics nutrient than the biotope of D-402.
-L334: I again consider that the use of “beneficial” is a misinterpretation of genomic evolution and connoted of too much finalism. The insertion of a transposon in the ALG67575 gene rendered the gene nonfunctional and the formation of sulphur globule impossible for D-401. This is probably why the D-401, unable to use sulfide as bioenergetics source, shifted to biotope with less sulfide. The comparison of D-401 and D-402T growth in presence of H2S or thiosulfate would have been very useful in this context.
Minor comments:
-L56: Please, correct “we studied of the difference” for “we studied the differences” or “we looked at the differences”.
-L67: Please, correct “was been increased” for “has been increased”.
-L125: please correct “second direction “ for “second dimension”.
-L168: please correct “had published” for “had been published”.
-L176: please note the genes (sqrQ, sqrF, fccB) all without capitals and in italic.
-L188: please correct “sulfite:ferricitochrom c” for “ferricytochrome c”.
-L233: please correct “but it does not to change” for “but it does not change”.
-L284: please correct “absence thiosulfate” for “absence of thiosulfate”.
-L298: please correct “in the gene encode sulfur” for “in the gene that encodes” or “in the gene encoding”.
-L344: please delete one “inserts” in “Inserts inserts”, and also in the title of the Table, in the SI.
Round 2
Reviewer 1 Report
No further comments.
Reviewer 2 Report
General comments:
The manuscript Microorganisms-828530-v2 entitled “Comparative genomics of Beggiatoa leptomitoformis 2 strains D-401 and D-402T with contrasting physiology 3 but extremely high level of genomic identity” by Rudenko et al. is a significant improved version of the original manuscript. I consider therefore this work acceptable for publication in “Microorganisms”.
Minor comments:
-L22: correct “…in contrast the D-402T…” for “…in contrast to the D-402T…”
-L196: please restaure “sorAB”
-L272: please correct “…to of B. leptomitoformis…” for ”… of B. leptomotoformis…”